# Health awareness and the transition towards clean cooking fuels: Evidence from Rajasthan

**Martina Zahno**[1], **Katharina Michaelowa**[1]*, **Purnamita Dasgupta**[2], **Ishita Sachdeva**[3]

**1** Department of Political Science, University of Zurich, Zürich, Switzerland, **2** Institute of Economic Growth, Delhi, India, **3** Department of Business Economics, Delhi University, Delhi, India

* katja.michaelowa@pw.uzh.ch

**Data Availability Statement:** The dataset and code required to replicate the study findings presented in this article are provided as Supporting Information in S6 Appendix.

## Abstract

Ensuring affordable, reliable, sustainable and modern energy for all by 2030 is part of the internationally agreed Sustainable Development Goals (SDG7). With roughly 3 billion people still lacking access to clean cooking solutions in 2017, this remains an ambitious task. The use of solid biomass such as wood and cow dung for cooking causes household air pollution resulting in severe health hazards. In this context, the Indian government has set up a large program promoting the use of liquefied petroleum gas (LPG) in rural areas. While this has led millions of households to adopt LPG, a major fraction of them continues to rely heavily on solid biomass for their daily cooking. In this paper, we evaluate the effect of simple health messaging on the propensity of these households to use LPG more regularly. Our results from rural Rajasthan are encouraging. They show that health messaging increases the reported willingness to pay for LPG, and substantially increases actual consumption. We measure this based on a voucher, which can only be used if LPG consumption is doubled until a certain deadline. Households exposed to health messaging use the voucher about 30% more often than households exposed to a placebo treatment. We further show that the impact of our very brief, but concrete health messaging is close to the effect of a 10% price reduction for a new LPG cylinder. Finally, our study raises some interesting questions about gender-related effects that would be worth consideration in future research.

## Introduction

In 2017, 2.9 billion people around the globe lacked access to clean cooking solutions [1]. Yet, ensuring affordable, reliable, sustainable and modern energy for all is part of the internationally agreed Sustainable Development Goals (SDG7). Simultaneously, the promotion of clean cooking contributes to a whole range of other SDGs [2]. Cooking with solid biomass such as wood and cow dung is a major source of household air pollution, which in turn is of one of the most important risks for global health, especially for women and children. Annually, millions of people die prematurely, with estimates for the years 2016 and 2017 ranging between 1.6 million [3] and 3.8 million [4]. Furthermore, cooking with solid biomass also causes major environmental burdens [5, 6] and impedes the empowerment of women and girls [2]. The

**Funding:** The project benefitted from a scholar exchange grant provided to K.M. and P.D. by the Indo-Swiss Joint Research Programme in the Social Sciences (Call 2015) jointly funded by the Indian Council for Social Science Research and the Swiss State Secretariat for Education, Research and Innovation (https://www.unil.ch/international/home/menuguid/evenements/archives/indo-swiss-programme.html). The funders had no role in study design, data collection and analysis, decision to publish, or preparation of the manuscript.

**Competing interests:** The authors have declared that no competing interests exist.

promotion of clean cooking thus contributes to gender equality (SDG5), climate action (SDG13), life on land (SDG15), and, most importantly, good health and well-being (SDG3).

In India, the health burden from traditional cooking is particularly high, with an estimated 482'000 premature deaths annually attributable to household air pollution [3]. As a consequence, in 2016, the Indian government started the "Pradhan Mantri Ujjwala Yojana" (PMUY) program to provide poor households with access to liquefied petroleum gas (LPG). As compared to combustion of traditional biomass, the combustion of LPG only generates a negligible amount of byproducts that are noxious to human health. As the PMUY covered the upfront cost for the access to this clean fuel, the number of households registered as LPG users increased by 80 million within just over three years. This is a development of unprecedented scale. If all these households fully switched to this clean cooking fuel, this would substantially increase the life expectancy in rural India.

However, so far, many households who adopted LPG under the program continue to rely on traditional biomass for a major part of their cooking. Multiple fuel use (so-called 'fuel stacking') is a widespread phenomenon and may persist over a long time [7–10]. A study based on multi-year LPG sales data from Karnataka shows that PMUY beneficiaries buy less than half the amount of LPG cylinder refills as compared to general consumers in rural areas [11].

There may be different reasons for this, notably the cost of regular refills and supply-side constraints [9, 10]. In addition, households may simply not be aware of the important health benefits of LPG and may thus not recognize any serious need for change. They usually see the time saving effects and the convenience of LPG, but this alone may not prompt them to switch to LPG as their primary cooking fuel. Hence, providing clear health information could be key to inducing the transition to sustained use of clean cooking fuels.

Related literature has shown that information on health benefits does not necessarily change people's behavior [12, 13]. Yet, to the best of our knowledge, this has never been tested in the concrete context at hand. The closest literature relates to LPG use in Kerala and Uttar Pradesh [14] and willingness to pay (WTP) for improved biomass stoves in rural Bangladesh [12], where upfront costs were relatively high and constituted an important barrier (see also [15]). This paper now examines the effect of health messaging in a context where the problem of high upfront cost is already taken care of through the PMUY. We expect that under these new conditions, the effect of health messaging on WTP and LPG consumption may be substantial.

Our evidence is based on a survey of 554 households in the rural part of Bikaner district in Rajasthan. We randomly assigned health information to one part of the respondents and general, non-health-related information on LPG to the other part. We then measured the treatment effect on two variables: (i) the necessary financial compensation to induce households to double their LPG consumption at given prices, and (ii) the actual increase in consumption, measured by the households' use of a voucher for a new refill before a given deadline. The two separate measurements allow us to distinguish the effect of health messaging from the potential nudging effect of the voucher itself. In our study we did not measure any direct health outcomes. However, the health benefits of regular LPG use are uncontested and do not depend on maintenance or the way the stove is used (which in contrast, would be highly relevant for an evaluation of improved biomass cookstoves).

Overall, our experimental evidence suggests that health messaging is highly effective and should be included in the campaigns to promote LPG. Our results show that even very brief and simple health messaging has a sizeable effect. We also discuss why it may be useful to target not just women, but also men. Moreover, our results confirm that without the additional health information, only very few people are aware of the severe health risks they incur by cooking with traditional biomass.

The remainder of this article is organized as follows: we first provide an overview of the extant literature regarding the effect of health messaging on household fuel choice and cooking habits. We then develop the conceptual framework and present the sampling strategy and methodological approach, including the experimental set-up. This is followed by the presentation of our results and a conclusion with policy insights from the study.

## Health awareness and fuel choices

While there is an extensive literature on household fuel choice decisions in low- and middle-income countries (see e.g. [7, 16, 17] for an overview), there are only a few studies examining the effect of health messaging on households' decision making in this respect. Furthermore, these studies focus on improved biomass cookstoves rather than LPG. What we can draw from the extant literature is that (1) generally, knowledge about the health hazards of traditional cooking is very limited, (2) the effect of health messaging seems to be context dependent, (3) the effect in the context of transition to LPG has hardly been examined yet, and (4) has not been examined at all in the context of the PMUY program implemented by the Indian government.

### Lacking knowledge about health hazards

Available evidence from several countries in Sub-Saharan Africa and South Asia (including India) suggests that the knowledge about the health hazards of traditional cooking in the affected population is very poor. While a majority of households recognize that there may be some health effects of household air pollution [12, 18], they largely underestimate the severity of these risks [12, 19]. When confronted with information about the fact that the adverse effects of household air pollution go way beyond transient irritations of eyes and throat, but substantially increase the risk of several deadly illnesses, they consider this information as highly salient [13].

The existing knowledge gap may be an important hindrance for the greater uptake and use of clean cooking fuels in India and elsewhere in the world [18]. To bridge this gap, the most natural intervention is to provide information (i.e., health messaging). Given that the knowledge gap is so profound, the effect of providing information may be substantial.

### No consistent evidence for the effect of health messaging

The few studies that have examined the effect of health messaging on household cooking fuel or technology use in a systematic way are primarily related to the introduction of improved biomass cookstoves. These stoves have often not been well accepted by households though, which is also true for India, where the adoption of improved biomass cookstoves has been limited [20]. Correspondingly, there is little evidence for any positive effect of health messaging on the willingness to pay for such cookstoves. In an experimental study in Uganda, only in one out of multiple settings do the messages increase willingness to pay, conditional on specific payment modalities. The direct reaction to payment modalities is much stronger [13]. Similarly, a broader study on fuel change in Myanmar finds no significant effect of health counselling [21], despite the fact that significant changes in the relevant knowledge are observed (see [22] for a review of these and other related studies). In contrast, findings from survey data from urban Indian households suggest that the belief that wood does not cause pollution significantly increases the quantity of firewood used [23]. Overall, the evidence suggests that the effect of health messaging depends on both technological features and financial constraints. Correspondingly, results from a survey-based study in China suggest that health messaging alone may not be effective unless it is coupled with access to improved technology in the form

of culturally well-adjusted stoves [24]. A randomized control trial in Bangladesh leads to simi-lar conclusions [25]. Other experimental studies highlight the importance of financial con-straints, notably in the form of liquidity constraints preventing the purchase of costly investment goods in Bangladesh and Uganda [12, 13]. This is in line with the fact that those studies providing the most convincing evidence for successful health messaging interventions tend to consider situations in which large upfront costs do not exist. They look beyond the purchase of new technology and focus on behavioral change such as taking children out of the kitchen or cooking outside or with open doors rather than in a closed room (see, e.g., [26]).

## Insufficient evidence in the LPG context

In the specific context of LPG use, there is almost no evidence on the role of health messaging. A notable exception is the field experiment by Krishnapriya that covers the effect of health messaging with respect to LPG uptake and multiple other household choices of fuels and appli-ances in rural communities of the Indian states of Kerala and Uttar Pradesh [14]. Households were confronted with information at different levels of intensity. It turned out that even the most intensive information campaign involving posters in the village, leaflets and one-to-one explanations to representatives of each household did not lead households to switch towards cleaner fuels, except for Kerala in those cases in which the information was provided to women. In contrast, with regard to electric appliances (such as the purchase of LED bulbs) the information treatment led to significant results. As purchasing a bulb requires a much smaller investment than purchasing an LPG stove, an explanation of the different reaction could be that LPG uptake is largely determined by liquidity constraints or financial constraints more broadly. This would be in line with the findings on other interventions discussed above.

## New situation as upfront costs are covered

If the above reasoning is correct, the introduction of the government's PMUY program in 2016 should have significantly changed the situation by removing the major constraint for the spread of the LPG technology. By offering LPG connections to poor rural households, the large upfront cost is taken care of by the government. 'LPG connection' hereby refers to the establishment of a formal account with a distributor as well as to the actual connection of the LPG stove to the LPG cylinder with a hose and a regulator. At current prices, this LPG connec-tion comes to 1600 INR (about 25 USD), and the additional cost for the first cylinder and the LPG stove is about 480 INR and 1020 INR respectively, i.e., another 1500 INR. Total upfront cost hence amounts to around 50 USD, which is difficult to bear for poor rural households. In the context of PMUY the government completely takes over the cost of the LPG connection, and in addition, it provides the opportunity to purchase the first cylinder and the LPG stove on the basis of an interest-free loan granted by the distributors that is gradually repaid by an increase in the price of subsequent refills by approximately 170 INR. Since no repayment is required if further refills are not purchased, households generally consider the initial uptake of the new technology as free of charge.

As mentioned in the introduction, this opportunity to receive an LPG connection, stove and the first cylinder initially free of charge has already driven 80 million households to adopt the new technology. If financial constraints are no longer binding, health messaging could now have a substantial impact on actual LPG use.

Of course, for a poor rural household, even the purchase of a refill for 480 INR (or more if some of the loan costs for the stove and the first cylinder are added to the bill of the refill) still represents some investment. Hence, financial constraints that prevented the uptake of the technology before the PMUY may still prevent some of the poorest households from

purchasing refills for the LPG cylinder. The fact that domestic LPG consumption has been growing at a much lower rate than what could be expected from the huge increase in connections may be a result of these remaining financial constraints [11, 27]. To what extent health messaging increases LPG consumption under the new financial conditions thus remains to be tested.

## Further drivers of LPG use

It is obvious that apart from our main variable of interest, i.e., health messaging, a host of other factors can be expected to influence the use of LPG. The energy transition literature, notably when it focuses on the use rather than just the adoption of clean cooking technologies, suggests those factors that may be relevant in our context [8, 10, 28]. This includes factors such as income and education, the opportunity costs of fuel collection [23, 28–30] and factors related to the social and cultural environment such as food taste preferences [31, 32]. In the context of the Indian PMUY program, there has also been a discussion whether or not LPG use may simply increase over time after the initial acquisition of the stove [11, 27, 33]. Variables such as time after adoption, income, education etc. can thus be useful to test the success of randomization and/or as controls when analyzing the effect of health messaging.

Finally, the literature suggests that there may be within-family differences in preferences for fuel choice, willingness to pay and reaction to health messaging, since women and children often lack decision power over financial issues but are at the same time most affected by indoor air pollution [12, 14, 34]. As discussed below, our empirical intervention has not been constructed to capture such effects. Yet, we will point at possible implications wherever this appears relevant.

## Conceptual framework

We propose an illustrative utility-maximization model to motivate and structure our analysis. Let us assume that a household derives its utility from energy services, which require cooking gas or other fuels as input, and from other goods. The different fuels used to provide energy services are imperfect substitutes as they also differ on a number of other utility-relevant factors, notably health. To arrive at a simple model that allows a focus on LPG, let the household's utility $U$ be defined as a function of the cooking gas LPG ($g$) and a composite good ($x$). The composite commodity $x$ represents the sum of all other consumption goods, including traditional biomass for cooking such as firewood. To ensure a certain degree of both complementarity and substitutability between $g$ and $x$ we use a standard Cobb-Douglas utility function:

$$U(g, x) = g^{\theta} x^{1-\theta} \tag{1}$$

with $0 < \theta < 1$.

The parameter $\theta$ captures the preference for cooking gas as compared to the composite good, and we assume that it is non-zero since all households we consider opted for an LPG connection through PMUY and stated that they intended to purchase a refill at some point in the future. For an income level of $B$ and prices $p_g$ and $p_x$, the budget constraint is given as:

$$g p_g + x p_x \leq B \tag{2}$$

Solving the optimization problem yields the Marshallian demand for LPG:

$$g^*(p_g) = \frac{B}{p_g} \cdot \theta \tag{3}$$

This equation shows the household's optimal LPG consumption as a negative function of the LPG price $p_g$. We can invert this function to obtain an expression of the price the household is willing to pay as a function of the amount consumed, for given preferences and budget:

$$p_g(g^*) = \frac{B}{g^*} \cdot \theta \tag{4}$$

The first (rather trivial) observation to note is that since $\frac{\partial p_g}{\partial g} < 0$, if a household is asked to consume more than $g^*$, the price it must pay will have to be reduced. What we are interested in here is how health messaging affects first, the discount a household demands when asked to consume substantially more LPG and second, the propensity of the household to actually increase consumption when provided with a pre-defined discount. Given the setting underlying our study, in which all households still have considerable leeway for more frequent LPG use, a consumption increase of 100% is used to indicate substantially higher consumption.

In the following, we first develop the theoretical expectations for the household's willingness to pay for LPG conditional on increased use, and then derive the predictions for the propensity to double gas consumption.

## WTP conditional on increased use

Let us consider that the preference for LPG $\theta$ is composed as follows: First, a basic preference $\bar{\theta}$ due to the convenience, time savings and other general benefits associated with cooking on the gas stove. Second, an additional appreciation based on the health benefits, reflecting knowledge of the health risks related to cooking with traditional biomass $h$, and the extent $\gamma$ to which this knowledge is salient for the decision maker. In particular, the salience $\gamma$ may vary based on the exposure to smoke from the traditional cookstove ("chulha"), and thus be higher for women than for men. Consequently, we define $\theta$ as:

$$\theta = \bar{\theta} + h \cdot \gamma \tag{5}$$

with $0 < h < 1$ and $0 < \gamma < 1$.

Now consider that we do not ask for the price households are willing to pay for their currently optimal—but very limited—consumption, but for a substantially increased consumption, namely a fixed $\bar{g} = 2g^*$. Including the specification for $\theta$ and this fixed consumption requirement into Eq (4), we obtain:

$$p_g(\bar{g}) = B\bar{g} \cdot \theta(h, \gamma) = B\bar{g} \cdot (\bar{\theta} + h \cdot \gamma) \tag{6}$$

Taking the derivatives of $p_g(\bar{g})$ with respect to $h$ and the cross-derivative with respect to $h$ and $\gamma$ provides us with the relevant theoretical predictions $\frac{\partial p_g}{\partial h} > 0$ and $\frac{\partial^2 p_g}{\partial h \partial \gamma} > 0$ (for computational details, see S1 Appendix). We thus expect to find two effects: (i) a positive effect of the health messaging on the price the individual is willing to pay for LPG, and (ii) a positive interaction effect with salience, i.e., in particular, a greater effect of the health messaging on women than on men.

Equivalently, we can express these hypotheses in terms of the compensation required by the individual to increase LPG consumption from $g^*$ to $\bar{g}$. The necessary compensation ($C$) corresponds to the market price of an LPG refill ($p_m$), which we can consider as given for our period of study, minus the price the individual is willing to pay ($p_g$), i.e., $C = p_m - p_g$. Hence, the effects with respect to the necessary compensation ($\frac{\partial C}{\partial h}$, and $\frac{\partial^2 C}{\partial h \partial \gamma}$) correspond to the above derivatives of $p_g(\bar{g})$ multiplied by $(-1)$.

While our experiment allows us to test the effect of $h$, the evidence we can provide on $\gamma$ is suggestive only, since our experiment was not designed to examine heterogeneous gender effects. This will be discussed further in the empirical part.

### The propensity of doubling gas consumption

Let us now consider actual change in consumption. This change can be observed through the use of a price-reducing voucher until a pre-defined, household-specific deadline. More precisely, the outcome variable of interest is the propensity of the household to use a voucher that ensures an increase of LPG consumption from $g^*$ to $\bar{g}$, for any level of a randomly determined price reduction $D$ and the resulting offer price $p_d = p_m - D$ specified on the voucher.

Let us denote voucher use by the indicator variable $Y$. Whether or not the voucher is used depends on the difference in utility $\Delta U$ between a situation in which the voucher is used $U_1$ and a situation in which it is not used $U_0$:

$$Y = \begin{cases} 1 & \text{if } \Delta U > 0 \\ 0 & \text{if } \Delta U \leq 0 \end{cases} \tag{7}$$

The difference in utilities itself reflects the (unobserved) propensity of voucher use. Taking into account the conditions for voucher use, namely doubling initial consumption and the discounted price $p_d$, $\Delta U$ can be expressed as:

$$\Delta U = U_1 - U_0 = \bar{g}^{\theta}(B - \bar{g}p_d)^{1-\theta} - g^{*\theta}x^{*1-\theta}. \tag{8}$$

To predict how the propensity to use the voucher will be influenced by health messaging, and how this in turn is affected by the salience of this information for the decision maker, we again compute the derivative with respect to $h$ and the cross-derivative with respect to $h$ and $\gamma$. This yields the relationships: $\frac{\partial \Delta U}{\partial h} > 0$ and $\frac{\partial^2 \Delta U}{\partial h \partial \gamma} > 0$ (see S1 Appendix).

The model thus predicts that, just like the WTP, the propensity to use the voucher (and hence the propensity to double consumption) should be positively affected by health messaging, and that this effect should again be greater for decision makers for whom the health information is more salient, namely for women.

## Empirical analysis

### Sampling and survey implementation

We tested our hypotheses in the rural communities of Bikaner, a district in the state of Rajasthan in Western India (see also the map in S2 Appendix). The selection was purposive as it fulfilled several criteria: Rajasthan was one of the first states to experience the launch of PMUY in May 2016. This means the program had been in its implementation phase for more than one year before the beginning of our survey in October 2017, so that the number of beneficiaries was sufficiently high for our sampling purposes.

Furthermore, available statistics on fuel use indicate that the district is quite representative for other parts of rural India (see Table 1). In 2011, 13% of the rural population in Bikaner district used LPG as their main cooking fuel as compared to 11% in rural India as a whole. There are only some differences regarding the choice of solid fuels that are used as an alternative. Given its dry climate and the related lack of vegetation, firewood is used less frequently in Bikaner than in other parts of India, while dung cakes are used more often. With respect to more general poverty-related indicators that may be relevant to fuel choice, Bikaner varies around the country average, with some factors above, and some factors below the all-India

**Table 1. Energy access and demographics, Bikaner vs. India 2011.**

|  | Bikaner | | India | |
| --- | --- | --- | --- | --- |
|  | **Total** | **Rural** | **Total** | **Rural** |
| LPG main cooking fuel | 29% | 5% | 29% | 11% |
| Firewood main cooking fuel | 53% | 75% | 49% | 63% |
| Dung cake main cooking fuel | 14% | 16% | 8% | 11% |
| Electricity for lighting (%) | 59% | 40% | 67% | 55% |
| Average literacy | 65% | 61% | 74% | 69% |
| Sex ratio (women per 1000 men) | 905 | 903 | 943 | 949 |
| Net domestic product p.c. (INR) | 52263 |  | 53331 |  |

Sources: [36–39]

average. For instance, per capita income is almost equal to the national average and access to electricity is higher in Bikaner than in the rest of India, while literacy rates are lower than average. The sex ratio is clearly below the Indian mean, which suggests that the status of women in the region is rather low (for a discussion, see, e.g., [35]). There is, however, a general North-South divide with respect to this indicator, and the rate we find for Bikaner district is close to the rates for the large Northern Indian states [36].

The sample consists of 554 households who received an LPG connection under the PMUY programme, but remained infrequent users. 55 villages were sampled from the census lists [36] with probability proportional to population size. For each village, a simple random sample of ten households was drawn from the village lists of PMUY beneficiaries. On average, there were 133 PMUY beneficiaries living in each village in the sample. Power calculations and the sampling procedure are described in S2 Appendix.

The sampling strategy with many villages and relatively few households within each village was chosen to ensure that all interviews could be run in parallel so that spillover effects would be minimized. Households that were unavailable, impossible to trace or that turned out to be ineligible for our sample were dropped and replaced from a back-up list of replacement households at the time of the first visit to the village. No repeat visit was made to a village.

Within each household, the preferred respondent was the main cook, who is usually a female. However, men were accepted as respondents if the relevant women were unavailable or unable to communicate to the enumerators for cultural reasons. Eventually, there were about 10% male respondents in the sample (see Table 1 in S3 Appendix). At the outset, preliminary screening questions were asked as follows:

1. Is the household indeed a PMUY beneficiary?

2. What is the frequency of use of cooking gas (LPG)?

These initial questions allowed us to screen out households that did not fit our criteria for infrequent use. We defined the corresponding threshold at a yearly LPG consumption of less than six cylinders a year for a family of five (excluding toddlers). An average Indian family using LPG exclusively requires 10-12 cylinders per year [11]. Thus, all PMUY households consuming less than 1.2 standard-size (14.2-kg) LPG cylinders per capita (for persons of age six and above) per year are considered infrequent consumers. Households covering all energy needs for cooking with LPG generally have a 50-100% higher consumption in the sampled villages [40].

The responses to these questions were verified by checking the entries in the respondents' official gas passbooks that report the households' average LPG consumption per year and the date of purchase of the cylinder currently in use. This information allowed us to compute the expected time until the next refill would become due based on past consumption patterns.

It should be noted that a number of initially selected villages and individual households had to be replaced in the sample: First, for some of the originally sampled villages, we were unable to obtain the list of PMUY beneficiaries. Second, in some villages, a very large number of households could not be traced as villagers were away for agricultural operations and had moved into so-called 'dhani', i.e., shelters in the fields scattered around the village. When this number became very high (over 30%), the whole village was replaced. Third, certain villages close to the India-Pakistan border were replaced due to security concerns. Eventually, the survey covered a total of 554 individuals from 55 villages.

Between September 2016 and March 2017 we carried out team building activities, some initial training of enumerators, a focus group discussion, pilots and key informant interviews to understand the situation on the ground and to refine our survey instruments. Subsequently, we established the cooperation with LPG distributors, requested the PMUY lists and analyzed secondary data sources from the Census and the National Sample Survey (NSS) as relevant for our sampling procedure. In October 2017 we conducted a final one-week intensive training workshop for the enumerators. The training included sessions on the rationale of the research design, exercises of the interviews including the implementation of the WTP-elicitation mechanism and the presentation of the different frames for the experiment (see below). It also included a familiarization of the enumerators with the use of the survey application 'Qualtrics' that allowed them to directly register all answers on electronic devices like tablets or smartphones. Based on this training, the enumerators—a team of students from Bikaner Agricultural University—carried out the data collection between October 2017 and February 2018. All household interviews were conducted in Hindi or Rajasthani (Marwari).

The survey had several domains. The first section inquired extensively on household demographic and socio-economic characteristics while the second part focused on specific questions to understand cooking and fuel use patterns. Subsequently, the survey application randomly assigned the health information to the households (probability = 50%), while the others received some general information on LPG supply and its characteristics. Following this, the enumerators assessed the required compensation for an increased use of LPG. Finally, the survey included several questions to test whether the respondents understood the health information provided.

The experimental set-up and the mechanism used to obtain the value of the required compensation are described in detail below.

## Experimental set-up

The intervention consisted in verbal information on the effect of traditional cooking on child development and diseases such as lung diseases, heart diseases and eye diseases. The enumerators were given a pre-formulated one-page text on these issues that they familiarized with and memorized in advance, so that they would keep their wording very close to the text without directly reading it out. The duration of the presentation of health hazards lasted for three to five minutes.

Given the possibility that any frame—or simply the time spent on talking about LPG—may affect the answers of the respondents [41], we constructed an alternative non-health related (and in this sense 'neutral' or placebo) frame for the control group. This frame consisted of

information on how cooking gas is extracted or produced from crude oil and then distributed to the households. The time spent on the information was similar for both frames.

To illustrate the verbal information, the enumerators carried along colored plasticized picture cards (size A4). We selected images that would be as neutral as possible while visually clarifying the spoken text. An English translation of the pre-formulated texts for both treatment and control group as well as a copy of the corresponding picture cards are presented in section 2 of S4 Appendix.

By design, the comparison of households who receive the health message and households who receive the placebo treatment reflects the net effect of the health messaging. If communicating about LPG over a certain time indeed has an effect by itself, the gross effect of health messaging (encompassing the effect of both the health-relevant content and the time of the LPG-relevant communication) should, in fact, be larger. As a consequence, our estimates of the treatment effect can be considered as a lower bound of the effect of health messaging for a population that would otherwise receive no LPG-related information at all.

After exposing the respondents to either of the two frames (health and non-health), we first assessed the households' stated WTP for LPG conditional on increased use and then observed households' actual consumption behavior by monitoring effective voucher use. Details on the measurement of these outcome variables are provided in the following sections.

## WTP conditional on increased use

There are several procedures used in experimental economics to measure willingness to pay in a way that ensures that rational individuals will reveal their genuine preferences. We base our WTP assessment for LPG on the Becker-DeGroot-Marschak (BDM) mechanism [42], a widely used option that mimics a Vickrey auction by replacing the other buyer with a random number. Under a common version of the BDM method, the person states a *bid* (for a good to purchase). The bid is then compared to a randomly determined *offer price*, that is, the price at which the good is made available to the bidding person. If the person's bid is higher than the offer price, the item is sold at the offer price. If the bid is below the price, no transaction happens and no payment is made. In this context, revealing one's true willingness to pay through the bid is a strictly dominant strategy.

A study on willingness to pay for water filters in northern Ghana demonstrated that the mechanism can be usefully applied even in contexts of low numeracy among the respondents [43]. To ensure that our respondents really understand the process, we explained each step of the procedure and followed it up by carrying out two rounds of the BDM mechanism with unrelated goods, first with a piece of soap, and then with a lighting bulb. If the respondents' bid was higher than the offer price, they paid the offer price and received the goods. Hence, by the time the respondents reached the LPG assessment, they were quite familiar with the procedure and had experienced that the implications of their decisions were real and binding.

With respect to LPG, the implementation of the BDM mechanism required adjustments due to the specific context of the study. First, real transactions with LPG cylinders are not possible, since LPG supply regulations in India imply that households can only purchase the refill from official distributors of oil marketing companies, and that, too, only once they have used and returned their empty cylinder. Hence, instead of concluding the transaction by selling an LPG cylinder at the reduced offer price to successfully bidding respondents, we handed out vouchers for the purchase of the next cylinder.

Second, we aim to elicit the WTP for LPG not as a good used only rarely for special occasions, but on a more regular basis, i.e., under the condition of *increased use*. This cannot be achieved simply by providing households with the offer to buy an additional cylinder. As our

sample only includes households that plan to buy a refill at some point over an infinite time horizon, all of them should be willing to purchase one at the market price $p_m$.

To obtain the relevant information on the WTP for increased consumption, the additional LPG use must be observable during a pre-specified period, i.e., before a certain deadline. As mentioned earlier, we chose a deadline relative to current use. More specifically, we fixed a specific deadline for each household that would require this household to consume the remainder of the LPG in the cylinder currently in use twice as quickly than under normal circumstances (see S4 Appendix for calculation details). The expiry date was clearly communicated to the respondent and written on the voucher. We also monitored that it was respected by the distributors.

We thus asked the respondents to make their bid for a new LPG cylinder under the condition of using up their current cylinder until the deadline. This bid was then compared to the randomly drawn discounted offer price $p_d$. The corresponding discount $D$ over the market price $p_m$ of 480 INR was designed to fall in the interval between 5 and 235 INR. Larger discounts were not expected to be necessary. The offer price itself was then between 245-475 INR and drawn from number cards in front of the respondents (for details, also on the choice of the price range, see S5 Appendix).

When the respondents stated a WTP which was at least as high as the offer price, and hence the (offered) discount ($D$) was greater than or equal to the required compensation for the increased use of LPG, they received the voucher, and they knew that they were expected to buy the next cylinder before the expiry date indicated on the voucher (see full protocol of WTP elicitation mechanisms as well as details on vouchers in S4 Appendix).

Unlike in the prior examples with the soap and the light bulb, we could, however, not enforce the final sale. This violates the conditions of the BDM mechanism because stating a bid that reveals the true required compensation is then no more a strictly dominant strategy. Indeed, it does no harm to consumers to make a higher bid since if they bid high enough to get the voucher, they do not need to actually make use of it. At the same time, it does not make them any better off to place a higher bid than the one that corresponds to their genuine willingness to pay. Hence revealing the truth remains a weakly dominant strategy.

In any case, a rational respondent will never make a bid that is too low. If at all, WTP will hence be overestimated by the procedure we chose. This may add to the effect we could obtain due to the fact that people under both the health and the alternative frame were confronted with some discussion on LPG (see above). For both reasons, average WTP obtained in our survey can be considered as an upper bound of the respondents' true WTP.

Note that the estimate of the health messaging effect should not be biased due to the enforcement problem. This is because there is no reason to believe that it might affect the treatment and the control group in different ways.

## Increase in LPG consumption

In the second part of our empirical analysis, we compare the actual voucher use by the households in the treatment and in the control group. Since the vouchers could be used only until the expiry date, the use of the voucher implies that the household truly consumed the remaining LPG in their current cylinder more quickly than usual, and that the incentive of the discount on the next cylinder was sufficiently strong to trigger this behavioral change. In addition, actual voucher use provides some insights into the sustainability of the initial impression made by the health messaging.

Two distinct factors should be considered in this context: First, while most of the time, the health information is only transmitted to the female respondent who also provides the

statement on WTP, the choice to double LPG consumption or not is the result of an intra-household decision-making process involving several household members. The actual purchase is usually carried out by men. These male family members (i) do not directly obtain the health message and (ii) will usually be less smoke-exposed than their spouses. Unless the information is transferred within the family very convincingly, this should reduce the effect of health messaging. Furthermore, the effect of health messaging should depend on the power of the respondent within the intra-household decision-making process.

Second, over time, the impression made by the health messaging may simply fade away. In the most extreme case, the information could be fully forgotten, in which case the intervention would have a zero effect on voucher use. In contrast, sharing health information and discussing it among family members may also increase its influence on the purchasing decision due to further reflection upon the topic, and respondents may develop a stronger preference for LPG when they are continuously exposed to the toxic smoke from the chulha after having learned what this exposure implies for their health. Depending on which of these causal channels dominates, health messaging may have a stronger or weaker effect on actual consumption behavior. The effect may also be stronger or weaker than what the respondents' immediate reaction measured in terms of their WTP may lead us to expect.

## Results

In a first step, we test whether our randomization allows us to successfully split the sample into two groups that are similar in all aspects that could be relevant for WTP and voucher use. Table 1 in S3 Appendix compares the means of both groups for a number of variables including socio-economic characteristics such as the respondents' age, education, religion, household size, their social category and proxies of income and wealth such as assets and land ownership (see S6 Appendix for data and code required to replicate all study findings reported in this article). Further variables describe the households' fuel choice and cooking behavior and capture preferences for and access to LPG: The average consumption of LPG, distance to the LPG sales point (zero in case of home-delivery), perceived convenience of LPG, knowledge about LPG subsidies and stated barriers to regular LPG consumption such as high refill costs or safety concerns. Finally, there are variables directly related to current LPG use and the conditions under which respondents were bidding, such as the number of days until the voucher's expiry date (voucher validity) and the content of the current cylinder at the time of the survey. Across all 25 variables, none of the differences in means is statistically significant at the 10% level. This implies that potentially confounding factors are well balanced across the two experimental groups.

The same holds if we limit the sample to those respondents who obtained a voucher (see Table 2 in S3 Appendix): Apart from a small difference in the share of Hindus and Muslims, the two experimental groups only differ with regard to WTP for LPG, which is a desired effect of our intervention. A description of all variables and summary statistics are provided in Table 3 in S3 Appendix.

### Impact on willingness to pay

Given the successful balancing of potentially confounding variables, we can now compare WTP for the treatment and the control group. Overall, the health messaging leads to an average increase in WTP of about 10 INR (from 352 to 362 INR, see Table 2 below).

The effect is not large, but the intervention was only very short and carried out by enumerators that were strangers to the respondents. Under conditions of more sustained health messaging by trusted health workers or members of the local community, the effect might have

**Table 2. Treatment effect on WTP, including controls.**

|  | (1) | (2) | (3) | (4) |
|---|---|---|---|---|
| Health message | 10.237* | 13.777** | 12.175** | 13.166** |
|  | (0.065) | (0.013) | (0.036) | (0.046) |
| Male |  | 31.863** | 54.111*** | 42.714** |
|  |  | (0.014) | (0.001) | (0.014) |
| Health message X Male |  | -41.385* | -62.277** | -53.083* |
|  |  | (0.072) | (0.020) | (0.068) |
| Voucher validity |  |  | -0.283 | -0.298 |
|  |  |  | (0.178) | (0.199) |
| Content |  |  | 4.245 | 5.845 |
|  |  |  | (0.766) | (0.698) |
| Asset index |  |  |  | 0.342 |
|  |  |  |  | (0.906) |
| Land |  |  |  | 15.647** |
|  |  |  |  | (0.029) |
| LPG distance |  |  |  | -0.170 |
|  |  |  |  | (0.673) |
| Fin. restriction |  |  |  | -14.355 |
|  |  |  |  | (0.172) |
| Education |  |  |  | 3.402 |
|  |  |  |  | (0.289) |
| Age |  |  |  | -0.341 |
|  |  |  |  | (0.316) |
| Household size |  |  |  | -0.853 |
|  |  |  |  | (0.574) |
| Months since LPG adoption |  |  |  | -0.191 |
|  |  |  |  | (0.645) |
| Constant | 351.678*** | 348.846*** | 352.230*** | 366.083*** |
|  | (0.000) | (0.000) | (0.000) | (0.000) |
| N | 539 | 539 | 468 | 455 |
| Adj. $R^2$ | 0.003 | 0.008 | 0.017 | 0.019 |

* $p < 0.1$,

** $p < 0.05$,

*** $p < 0.01$. $p$-values based on standard errors clustered at village level in parentheses. For the additional variables in Col. 3 and 4 complete data is not available for the full sample, resulting in a smaller number of observations.

been much stronger. Furthermore, remember that the estimate reflects the net effect of health messaging, and that the gross effect could be larger if the time of the communication on LPG has a positive effect by itself.

To provide some more detail on this result Fig 1 displays the cumulative distribution of the respondents' stated WTP in both treatment groups.

The share of respondents that accepted prices in the upper half of the price range is consistently higher among subjects who were confronted with the health information. Fig 1 also shows that the estimated median WTP is at 350 INR per cylinder. Since these estimates must be considered as an upper limit of the true WTP of our respondents (see Conceptual framework), they are well in line with the results of an earlier large-scale household survey in six Indian states, which suggest that households who are interested in adopting LPG would be

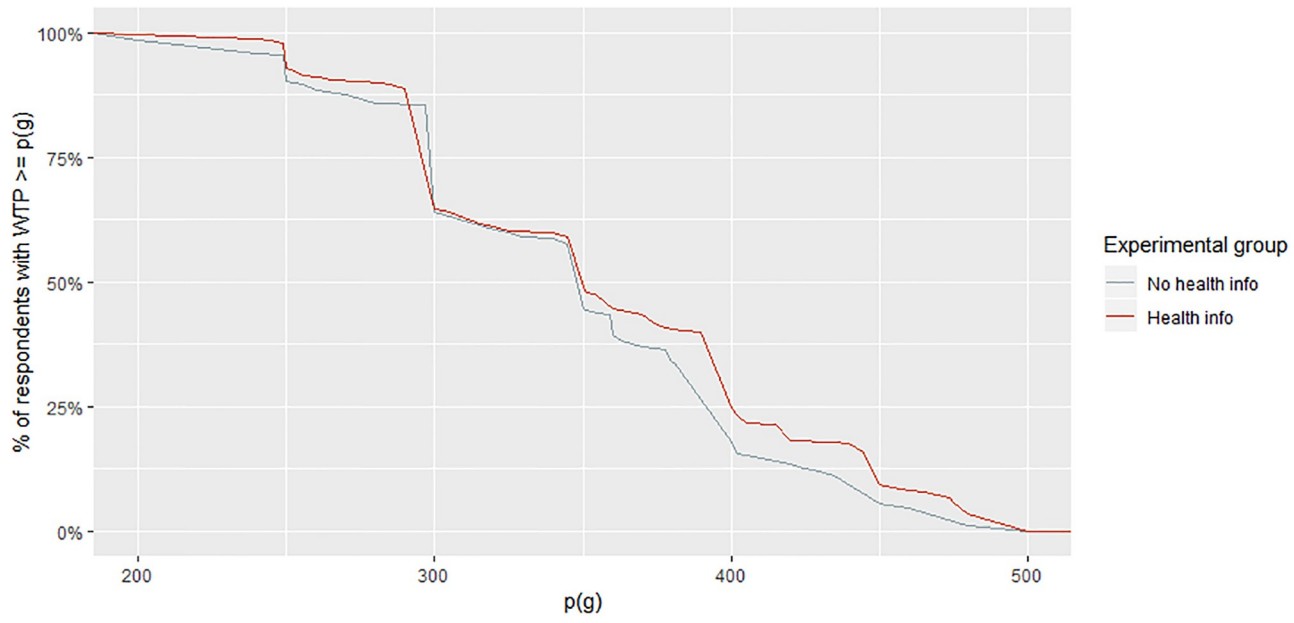

**Fig 1. WTP for LPG conditional on increased use, by experimental group.**

ready to pay 300 INR per month (median) to cover all cooking needs with LPG [27]. While health messaging increases WTP, substantial additional subsidies will still be required to induce poor households to become more regular LPG consumers.

Apart from the direct effect of our intervention and the respondents' price-elasticity of demand, their stated WTP may be influenced by some additional factors. In particular, our previous discussion suggests that gender differences due to differences in smoke exposure and time spent cooking should affect the impact of health messaging. It may also be relevant to control for the content of the current cylinder and the time left until the voucher needs to be used. Few households had a full cylinder at the time of the survey, such that the requirement to speed up consumption referred to different absolute quantities. Households might agree to a lower compensation when their cylinder is already partly used. Similarly, the absolute time period over which the behavioral change to double LPG consumption is required varies between households. Due to differences in the usual speed of consumption, this may be true even if the filling of the cylinders is initially the same for two households. In principle, this could have implications for WTP, too. For instance, households might feel that a behavioral change over a small period of time—maybe just a few days—is easier to achieve than a change over many weeks. In this case, WTP should be higher if the survey happens closer to the date at which the next refill would have been required anyway. However, one could also imagine that having more time enables the household to plan the increased consumption in a better way, i.e., by using LPG rather than the chulha when many guests are in the house, which may not happen that frequently. Moreover, time preference would imply that a compensation to be received in the far away future would be valued less than a payment one could receive within a few days.

Whether these considerations do affect the respondents' stated WTP, and if so, in which direction, will be examined below. We will also add further controls for potentially relevant household characteristics.

**Multivariate regression analysis for WTP.** Table 2 presents the results with Col. 1 as a baseline, a dummy variable for male respondents and its interaction with the treatment variable in Col. 2, and a number of key control variables in Col. 3 and 4.

While Col. 1 shows the already discussed effect of the intervention without any other variables, in Col. 2 we distinguish between the effects for male and female participants of our survey. As mentioned already, there are only few male respondents in our sample and they systematically differ from average men in the communities of interest. In most cases, these men belong to very traditional families as they did not allow their spouses to talk to the enumerators. This also suggests a highly unequal balance of power in these households. Our sample therefore does not allow us to provide general estimates for heterogeneous treatment effects between women and men. Yet, it is important to understand to which extent the specific households in which we interviewed men rather than women are different, and thus affect our results. Furthermore, the differentiation by gender within our sample can provide some suggestive evidence as a basis for the analysis of gender effects in future research. These considerations lead us to systematically present whatever suggestive results we have, calling for verification in future studies.

Note that due to the inclusion of the interaction term, the coefficient estimate for health messaging now refers to female respondents alone. With a point estimate of about 14 INR it is higher than the average for all respondents (male and female) in Col. 1. Correspondingly, the negative coefficient of the interaction term suggests that men in our sample react to health messaging much less than females. In other words, at least for the specific selection of male respondents in our sample, results are in line with our expectations. The main effect of the dummy for male respondents further indicates that within our sample, men generally state a much higher WTP than women. This seems to be a common result for WTP assessments in households in which women are not used to committing to major payments, and does not specifically relate to LPG (see also [13]). In our sample, only 3% of women report taking decisions on the purchase of durable goods on their own.

Col. 3 then adds controls for the remaining content of the cylinder at the time of the survey, and the period of validity of the voucher. Neither of the two are significant. This suggests that neither variable plays a major role in determining the compensation for increased consumption requested by the household. Rather, households seem to just consider the required change in behavior in terms of the relative increase in consumption, no matter the period over which this change is requested, and no matter what absolute quantity of LPG consumption this implies.

Col. 4 further adds a number of other control variables that might be relevant for the willingness to pay. The only significant variable is the dummy for land ownership, reflecting that wealthier households owning some land tend to have a higher WTP. Note that the effects of the control variables (or the lack thereof) should not be over-interpreted as some of them are highly correlated. They are included mainly to show the robustness of the main results. The estimated treatment effect remains positive and significant throughout (with little change in size across Col. 2-4 in which it refers to females).

## Impact on voucher use

So far, the results thus confirm our hypotheses. But is the voucher that allows the household to buy the next LPG cylinder at or below the price of reported WTP actually used? Does it indeed lead to the requested behavioral change of doubled consumption until the given deadline? Overall, in 303 out of 539 conducted BDM procedures, the respondent's bid (i.e., stated WTP) was sufficiently high to receive a voucher. The voucher values, i.e., the discounts offered on the

purchase of the next LPG cylinder, range from 5 to 235 INR, and about 70% of them lie above 150 INR. For 296 vouchers handed out to households we could trace whether the beneficiary had used the voucher to cover a part of the household's next LPG purchase.

It turns out that only 35% of these 296 households actually used the voucher. Unfortunately, we are unable to disentangle the different possible reasons discussed above, and any combination of these could be responsible for this result. What we can examine, however, is the extent to which our intervention, namely the health messaging, affected the actual use of the voucher.

**Effect of health messaging on voucher use among voucher owners.** Given the binary nature of the dependent variable, we proceed with logistic regressions presenting both odds ratios and predicted probabilities. For comparison, S6 Appendix further includes the code for linear probability models, which show very similar results. Just as in our analysis of WTP, we first present a simple bivariate estimation of the treatment effect and progressively add more variables.

Based on the 296 available observations, we find a strong and significant effect of health messaging on voucher use. Table 3, Col. 1 indicates that the odds of using the voucher are 1.63 times higher for households that received the health message. This corresponds to an increase in the probability of voucher use by 11 percentage points, from 30 to 41% (see S6 Appendix). The treatment hence increases the probability of voucher use by more than one third.

The effect is even more remarkable given that quite some time passes between the treatment and potential voucher use, and given the required intra-household transfer of the information. In addition, since spill-overs between the treatment and control group cannot be avoided during the period until voucher use, this result represents a lower limit of the actual effect. Finally, as before, we should remember that we only estimate the net effect of health messaging, not including the possible impact of LPG-related communication time, which was the same for both treatment groups.

Col. 2-4 of Table 3 add a differentiation by gender of the respondent and further controls. Due to the interaction term with the dummy variable for male respondents, odds ratios shown for health messaging in these models refer to female respondents only. In contrast to theoretical expectations, it seems that the positive and significant overall effect is now primarily driven by the few male respondents in our sample. The interaction term itself, which shows how much the effect of health messaging differs between male and female respondents, is strong and significant in two out of three regressions. It indicates that for the male respondents, the odds ratio of health messaging is eight times as high as for the female respondents in our sample (Col. 2). Adding further controls somewhat reduces this difference, but the estimate remains sizeable. Furthermore, for men alone, the treatment effect is always positive and significant (not shown), while this is not the case for female respondents (cf. Col. 2). This is surprising since the male respondents in our sample initially did not seem to react to the treatment—as measured by their statement on WTP.

This suggests some interesting household dynamics after the visit by our enumerators. Due to their lack of power within the household, women might have more difficulties in transforming their initially voiced preferences into the household's final purchasing decision. Hence, even if their greater exposure to smoke leads them to react more strongly to health messaging in the first place, they may not always be in a position to actually push for a greater use of LPG. Men, once convinced, do not have this problem. At the same time, they seem to require more time to react to the health information received. They might first cross-check this information and/or discuss the issue within the family and with friends. This suggests that considering given power relations within rural Indian households, it is important to convince men about the health benefits of LPG, and not just women. Further research is required to examine heterogeneous effects of health messaging and intra-household decision-making on fuel choices.

**Table 3. Treatment effect on voucher use, including controls.**

| | (1) | (2) | (3) | (4) |
|---|---|---|---|---|
| Health message | 1.628** | 1.396 | 1.633* | 1.950** |
| | (0.047) | (0.198) | (0.083) | (0.029) |
| Male | | 0.885 | 1.004 | 1.569 |
| | | (0.826) | (0.995) | (0.515) |
| Health message X Male | | 8.379** | 5.948* | 4.280 |
| | | (0.030) | (0.087) | (0.190) |
| Voucher validity | | | 1.002 | 0.999 |
| | | | (0.880) | (0.963) |
| Content | | | 0.460 | 0.379 |
| | | | (0.312) | (0.228) |
| Asset index | | | | 1.076 |
| | | | | (0.473) |
| Land | | | | 1.231 |
| | | | | (0.504) |
| LPG distance | | | | 1.025* |
| | | | | (0.088) |
| Fin. restriction | | | | 1.702 |
| | | | | (0.127) |
| Education | | | | 1.012 |
| | | | | (0.931) |
| Age | | | | 0.981 |
| | | | | (0.325) |
| Household size | | | | 0.878* |
| | | | | (0.098) |
| Months since LPG adoption | | | | 1.038* |
| | | | | (0.055) |
| WTP for LPG | | | | 0.995** |
| | | | | (0.033) |
| Constant | 0.429*** | 0.435*** | 0.524** | 4.362 |
| | (0.000) | (0.000) | (0.030) | (0.226) |
| N | 296 | 296 | 254 | 247 |
| Area under the ROC curve | 56% | 58% | 62% | 72% |

Logit models with odds ratios,

* $p < 0.1$,

** $p < 0.05$,

*** $p < 0.01$. $p$-values in parentheses. Lack of data on the additional variables included in Col. 3 and 4 lead to a reduction in the number of observations.

Adding further control variables substantially improves the overall prediction of voucher use as indicated by the receiver operating characteristic (ROC) curve, notably in Col. 4. In this specification, the estimate of the treatment effect is even higher than before. Table 4 presents the results of this model in terms of predicted probabilities, across experimental groups and gender of the respondents. It indicates that health messaging increases the probability of voucher use by 16 percentage points overall, which reflects the 14 percentage point increase when the health messaging was delivered to women, and a massive 44 percentage point increase for the few cases in which the messaging was delivered to men.

**Table 4. Predicted probabilities of voucher use.**

|  | No health message | Health message | Difference | p-value | N |
|---|---|---|---|---|---|
| *Total* | 0.273 | 0.437 | 0.164 | 0.005 | 247 |
| *Females* | 0.266 | 0.401 | 0.135 | 0.026 | 225 |
| *Males* | 0.354 | 0.791 | 0.437 | 0.019 | 22 |

Estimates are based on the logit model presented in Table 3, Col. 4.

Regarding the individual control variables, there is no surprise. As before, many of them are insignificant, this time including the indicators we use for wealth and income. This is consistent with our model since household budget is as relevant for $U_0$ without the voucher as for $U_1$ with the voucher and hence cancels out for $\Delta U$ (see S1 Appendix). The significant control variables are distance from the sales point, household size, months since LPG adoption, and WTP.

The latter may deserve some additional explanation. The odds ratio for WTP is smaller than one, reflecting a negative relationship between WTP and voucher use. At first glance, this may seem unexpected. However, WTP is, by design, negatively related to the discount. Households obtain the vouchers only if their stated WTP is higher than the randomly drawn, discounted offer price. Hence, the higher this price (i.e., the lower the discount $D$), the higher their WTP must be for them to be included in the sample for the estimations in Table 3.

The latter also leads to a more general risk of selection bias, even when we control for WTP. The average WTP in the sub-sample of voucher owners is significantly higher than the WTP of those respondents who did not receive a voucher (390 vs. 314 INR). As a result, the sub-sample may not be representative of our initially drawn sample of typical PMUY users.

This problem also affects our estimate of the treatment effect. As the health messaging affects WTP, it also affects the selection into the sub-sample of voucher owners. Studying the treatment effect within this sub-sample will thus not provide us with a valid estimate for the full impact of our intervention.

**Joint effect of health messaging on voucher use.** In order to avoid the selection problem discussed above, we additionally estimate the *joint* effect of health messaging on voucher use. That is, we now use the total sample of respondents, no matter whether they obtained a voucher or not, and set the outcome variable "voucher use" to zero for those respondents who did not receive a voucher in the first place (as their WTP was below the randomly drawn offer price). The share of voucher users in the total sample of households in our sample is now 20% (among voucher owners only, it was 35%). We use a fixed effect for each offer price, as the chance to obtain a voucher with a given WTP increases with decreasing offer prices. The fixed effects will thus provide a substantial part of the explanation for the zero-values in the outcome variable.

Table 5 shows the results of logit models similar to those in Table 3. Without offer-price fixed effects, being confronted with the health message increases the odds of a household using a voucher (and thus demonstrating doubled consumption) by a factor of 1.44 (Col. 1). This corresponds to an increase in the probability of using a voucher by 6 percentage points, from 17 to 23% (see S6 Appendix). While the absolute value of the increase is smaller than in the sub-sample of voucher owners (6 as compared to 11 percentage points), in relative terms, the increase is thus as important as before. In both samples, the treatment increases voucher use by more than one third.

**Table 5. Joint effect of health messaging on voucher use.**

| | (1) | (2) | (3) | (4) |
|---|---|---|---|---|
| Health message | 1.444* | 1.616** | 1.504 | 1.942** |
| | (0.095) | (0.046) | (0.111) | (0.026) |
| Male | | | 1.484 | 2.450 |
| | | | (0.496) | (0.212) |
| Health message X Male | | | 2.383 | 1.479 |
| | | | (0.263) | (0.674) |
| Content | | | | 0.351 |
| | | | | (0.174) |
| Voucher validity | | | | 0.999 |
| | | | | (0.915) |
| Asset index | | | | 1.078 |
| | | | | (0.441) |
| Land | | | | 1.247 |
| | | | | (0.468) |
| LPG distance | | | | 1.027** |
| | | | | (0.049) |
| Fin. restriction | | | | 1.312 |
| | | | | (0.437) |
| Education | | | | 1.035 |
| | | | | (0.778) |
| Age | | | | 0.978 |
| | | | | (0.220) |
| Household size | | | | 0.870* |
| | | | | (0.056) |
| Months since LPG adoption | | | | 1.025 |
| | | | | (0.174) |
| Constant | 0.203*** | 0.461 | 0.428* | 0.922 |
| | (0.000) | (0.117) | (0.094) | (0.933) |
| N | 532 | 465 | 465 | 396 |
| Offer price fixed effects | No | Yes | Yes | Yes |
| Area under the ROC curve | 55% | 70% | 71% | 77% |

Logit models with odds ratios,

* $p < 0.1$,

** $p < 0.05$,

*** $p < 0.01$. $p$-values in parentheses. As some of the highest offer-prices perfectly predict failure to use the voucher, and as the additional control variables have some missing values, Col. 2-4 include a smaller number of observations.

By adding offer-price dummies in Col. 2-4, the overall quality of the prediction markedly increases as indicated by the area under the ROC curve. This reflects the relevance of the price effect. The estimate of the treatment effect also becomes more precise, and larger than before.

In terms of the differences between male and female respondents in our sample, the results point in the same direction as before, but the interaction term is insignificant (Col. 3). When including control variables in Col. 4, our results suggest that health messaging almost doubles the odds of voucher use among the households with female respondents (odds ratio = 1.94). The point estimate for the few male respondents is again even higher, but the difference

**Table 6. Predicted probabilities of voucher use.**

|  | No health message | Health message | Difference | p-value | N |
|---|---|---|---|---|---|
| *Total* | 0.167 | 0.267 | 0.100 | 0.011 | 396 |
| *Females* | 0.157 | 0.249 | 0.091 | 0.024 | 366 |
| *Males* | 0.287 | 0.490 | 0.202 | 0.215 | 30 |

Estimates are based on the logit model presented in Table 5, Col. 4.

remains insignificant. Table 6 shows these effects in terms of predicted probabilities, for the whole sample and by gender of the respondents.

While we have so far used fixed effects to account for differences in the offer price, we can also include the offer price $p_d$ directly as an explanatory variable. This also allows us to directly interpret the relationship between prices and voucher use (see S5 Appendix). In line with the predictions of our theoretical framework (S1 Appendix), we find that price reductions have a strong positive effect on the household's purchasing decision. Across all available observations, a price discount of 40 INR (8.3% of the current subsidized price of a new cylinder) is estimated to increase the probability of voucher use on average by about 10 percentage points. This corresponds to the estimated impact of health messaging.

The effect of health messaging remains relatively stable across the range of discount values, and hence, there is little evidence that the two measures interfere with each other (see Fig 2 in S5 Appendix). This implies that health messaging can be usefully topped up by price reductions to reach an even greater overall effect on demand.

## Testing the information channel

While we argue that the success of the intervention is based on the respondents' greater health awareness, this has not been directly tested so far. In this last part we will thus assess the effect of the health messaging on health-related knowledge. To that aim, we compare post-intervention responses of the treatment and the control group to several questions regarding the health hazards related to traditional cooking (see S4 Appendix for post-intervention questionnaire).

First, we examine the response to the question whether traditional cooking affects health slightly, severely, or not at all. Our dependent variable is a binary indicator of the belief that there are serious health hazards involved. Without any further information, respondents knew very little about health hazards related to smoke from the chulha, leaving much scope for improvement: under the alternative frame, only 13% believed that there were serious health hazards related to cooking with traditional biomass, while 60% believed that there were just some minor transitory effects, and 27% were of the opinion that there were no health effects at all.

This was also confirmed in complementary qualitative interviews with other households prior to the experiment. When women were asked about health effects, they primarily thought of these as temporary irritations such as coughs or watering eyes, and stated that these were not problems of any major consequence, but rather something to get used to over time. A comparison of our findings with previous surveys in India [18, 27, 40] hence underlines the importance of enquiring specifically about knowledge of *serious* health risks. According to a large survey among rural Indian households in 2018 [27], 84% of those who relied on biomass as primary cooking fuel stated that cooking with LPG is better than using a traditional cookstove regarding the health impact. This may be seen as indication that most households are

aware of adverse health impacts related to using biomass. While this share is comparable to the proportion of households who are aware of *some* (major and minor) health impacts in our sample, the results from our more detailed questionnaire demonstrate that this is merely superficial knowledge and that the vast majority of these families lack awareness of the *severe* health risks related to household air pollution.

In the context of such limited initial knowledge, health messaging led to a strong and highly significant increase of reported awareness of serious health hazards. Among respondents that received the health information, 48% report to be aware of serious adverse health effects, i.e., reported awareness is four times as high as before. The total share of individuals who reported to be aware of health risks (serious or less serious) increased to 94%.

Table 7, Col. 1 presents the results for severe effects, distinguishing by gender. Among female respondents, 12% of the untreated report that they are aware of severe health issues as compared to 46% (12+34%) of the treated. There are no significantly different responses for male respondents. All results are robust to the addition of further control variables (not shown).

Of course, the treatment effect of the health message may be partly due to social desirability bias: after the information treatment, respondents know that a positive answer is expected and might hence pretend to be aware of severe health hazards even without fully understanding or being really convinced.

We thus consider a second dependent variable, which requires concrete knowledge about health hazards incurred when using traditional solid fuels for cooking. This variable reflects the share of diseases related to indoor air pollution (IAP) correctly identified within a set of ten diseases out of which only six are indeed related to IAP. Col. 2 presents the results. They confirm those of the previous estimation. Our brief health message increases the share of correctly identified smoke-related diseases by 15 percentage points for female respondents—and similarly for male respondents since the interaction term is very small and statistically insignificant. Independently of the health messaging, in our sample, female respondents generally recognize diseases related to traditional cooking substantially better than male respondents.

**Table 7. Treatment effect on health-awareness.**

| | (1) | (2) | (3) |
|---|---|---|---|
| | Severe effects | IAP diseases | All diseases |
| Health message | 0.343*** | 0.150*** | 0.066*** |
| | (0.000) | (0.000) | (0.000) |
| Male | 0.132 | -0.176*** | -0.115*** |
| | (0.116) | (0.000) | (0.000) |
| Health message X Male | 0.157 | 0.029 | 0.023 |
| | (0.174) | (0.590) | (0.570) |
| Constant | 0.118*** | 0.280*** | 0.482*** |
| | (0.000) | (0.000) | (0.000) |
| N | 503 | 539 | 539 |
| Pseudo $R^2$ / Adj. $R^2$ | 0.140 | 0.096 | 0.084 |

Col. 1 shows average marginal effects based on a logit model, as the dependent variable is binary. Col. 2 and 3 show linear regression models.

* $p < 0.1$,

** $p < 0.05$,

*** $p < 0.01$. *p*-values based on standard errors clustered at village level in parentheses.

As a third dependent variable, we examine the share of correctly identified diseases among all ten diseases. A value of one on this variable implies that not only the IAP-related, but also the IAP-unrelated diseases are correctly identified. This ensures that high values on the dependent variables cannot be obtained simply by responding in a way that relates all kinds of diseases to cooking habits. Hence the values of this variable cannot be driven by social desirability bias. When using this variable, the treatment effect is smaller (only about 7 percentage points), but remains highly significant (see Col. 3). As before, the treatment effect does not differ between male and female respondents, i.e., there seems to be no difference regarding the capacity to absorb the health information we provide. But again, overall, women recognize the relevant diseases substantially better than the men in our sample. Unless the difference is driven by the particular selection of men among our respondents, this gives some plausibility to our expectation that women who are exposed to smoke on a daily basis may find the knowledge about smoke-related diseases more important than men. This could explain why they tend to be somewhat better informed already prior to our intervention.

In sum, the empirical evidence thus confirms that the intervention increases the respondents' knowledge about the health hazards related to traditional cooking. Despite some differences in initial knowledge, this is true for both women and men in our sample, with no observable difference in the treatment effect. This implies that the differences we observed between male and female respondents regarding the impact of health messaging on WTP and voucher use cannot be explained by differences in the capacity to absorb the information we provide. While additional research is required to confirm these results with a representative sample of women and men, this is in line with our theoretical argument, which suggests that gender differences should be driven by differences in the salience of the information rather than by the information itself. Of course, as we have seen, such differences may be overridden by practical constraints related to the limited power of females in household decision making over expensive items.

## Conclusion

Traditional cooking habits based on the use of solid fuels such as cow dung and firewood affect a range of SDGs. In particular, they generate severe health hazards. With an estimated 846 million people being exposed to household air pollution in India, the corresponding health burden is particularly high [3]. This paper examined to what extent health messaging for poor rural households can mitigate the problem. Based on a survey in rural Bikaner district (Rajasthan), we analyzed the effect of a health messaging intervention on willingness to pay and the propensity to consume more LPG, a clean fuel, which all of our sample households already have access to in principle through the Indian government's PMUY program.

Our results show that health messaging increases the reported willingness to pay for LPG, and leads to substantially higher actual consumption among households who currently use LPG only on a very infrequent basis. We measure this based on a voucher which can only be used if LPG consumption is doubled until a certain deadline. Households exposed to health messaging use the voucher about 30% more often than households exposed to a placebo treatment. We further show that the impact of our very brief, but concrete health messaging is as strong as a decrease in the price of a new LPG cylinder by about 40 INR.

Obviously, health messaging does not need to be considered as an alternative to price reductions. Our results confirm prior studies indicating that the willingness to pay for regular LPG use by a typical poor rural household is considerably below the current regulated market price of 480 INR per cylinder. It may thus be useful to combine health messaging and price

reductions. We find that these two measures do not interfere much with each other and can thus be decided upon independently.

Our results also confirm that the health messaging indeed increases the respondents' knowledge about smoke-related diseases, which is an important precondition for the causal effect we claim. It should be noted that without any health messaging, the relevant knowledge is extremely low. Among the untreated, only 13% of all respondents believe that cooking with traditional biomass entails any serious health risks. This percentage increases to 48% in the treatment group. The low initial knowledge may be one reason why we find such substantial effects on LPG use.

Our empirical estimation was not designed to estimate heterogeneous treatment effects between women and men. Nevertheless, our study suggests some potentially relevant, and partially unexpected gender differences that call for further investigation in future work. Independently of these results, given that women often lack decision-making power on major purchases, knowledge building should not target women alone.

## Supporting information

**S1 Appendix. Mathematical derivations for the theoretical model.**
(PDF)

**S2 Appendix. Power calculation and sampling protocol.**
(PDF)

**S3 Appendix. Summary statistics and balance tests.**
(PDF)

**S4 Appendix. Protocol and material for WTP elicitation and experiment.**
(PDF)

**S5 Appendix. Offer price distribution and voucher use.**
(PDF)

**S6 Appendix. Replication data and code.**
(ZIP)

## Acknowledgments

We are grateful for the facilitation of the research implementation by the Ministry of Petroleum and Natural Gas (Govt. of India), Indian Oil Corporation Ltd. and all concerned field managers and LPG gas distributors in Bikaner district. We also acknowledge the contribution to the data collection by Rajesh Sharma and a team of students at the SKRAU Agricultural University Bikaner. We thank Narendra K. Arora, Lorenzo Casaburi, Paula Castro, Ipsita Das, Sebastian Fehrler, Michael Grimm, Manish Grover, Robert Huber, Ashutosh Jindal, Lennart Kaplan, Stefan Klonner, Olexiy Kyrychenko, Jan-Walter De Neve, Jörg Peters, Massimo Phillipini, Atonu Rabbani, Alfonso Sanchez, Kirk R. Smith, E. Somanathan, Ashok Sreenivas, Johannes Urpelainen, and two anonymous reviewers for their insightful comments that have helped us to shape and improve this paper.

## Author Contributions

**Conceptualization:** Martina Zahno, Katharina Michaelowa, Purnamita Dasgupta.

**Data curation:** Martina Zahno.

**Formal analysis:** Martina Zahno, Katharina Michaelowa.

**Funding acquisition:** Martina Zahno, Katharina Michaelowa, Purnamita Dasgupta.

**Investigation:** Martina Zahno, Katharina Michaelowa, Ishita Sachdeva.

**Methodology:** Martina Zahno, Katharina Michaelowa, Purnamita Dasgupta.

**Project administration:** Martina Zahno, Katharina Michaelowa, Purnamita Dasgupta, Ishita Sachdeva.

**Resources:** Martina Zahno, Ishita Sachdeva.

**Supervision:** Katharina Michaelowa, Purnamita Dasgupta.

**Validation:** Martina Zahno.

**Visualization:** Martina Zahno.

**Writing – original draft:** Martina Zahno, Katharina Michaelowa.

**Writing – review & editing:** Martina Zahno, Katharina Michaelowa, Purnamita Dasgupta, Ishita Sachdeva.

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
