## [Decision Letter · Decision Letter 0]

3 Jan 2020

PONE-D-19-28805

Health awareness and the transition towards clean cooking fuels: Evidence from Rajasthan

PLOS ONE

Dear Prof. Michaelowa,

Thank you for submitting your manuscript to PLOS ONE. After careful consideration, we feel that it has merit but does not fully meet PLOS ONE’s publication criteria as it currently stands. Therefore, we invite you to submit a revised version of the manuscript that addresses the points raised during the review process.

The authors should address all issues raised by two anonymous reviewers. In particular, any conclusions about gender-effects should be closely considered given some of the inherent limitations of their experiment.  Although it was mentioned by one reviewer, there is no need to number subsections. However, the authors should consider a mode descriptive heading for 'Appendix 1'. Given the nature of this research the authors should be fully compliant by making all experimental (laboratory) protocols available.

We would appreciate receiving your revised manuscript by Feb 17 2020 11:59PM. To enhance the reproducibility of your results, we recommend that if applicable you deposit your laboratory protocols in protocols.io, where a protocol can be assigned its own identifier (DOI) such that it can be cited independently in the future. For instructions see: http://journals.plos.org/plosone/s/submission-guidelines#loc-laboratory-protocols

We look forward to receiving your revised manuscript.

Kind regards,

Francisco X Aguilar

Academic Editor

PLOS ONE

Journal Requirements:

[We are grateful for the support of the Indian Ministry of Petroleum and Natural Gas, 839

Indian Oil Corporation Ltd. and all concerned field managers and LPG gas distributors 840

in Bikaner district in the implementation of this research. We also acknowledge the 841

contribution to the data collection by Prof. Rajesh Sharma and a team of students at 842

the SKRAU Agricultural University Bikaner. We gratefully acknowledge a Scholar 843

Exchange Grant awarded by the Indo-Swiss Joint Research Programme in the Social 844

Sciences jointly funded by the Indian Council for Social Science Research and the Swiss 845

State Secretariat for Education, Research and Innovation. Finally, we thank Lorenzo 846

Casaburi, Paula Castro, Ipsita Das, Jan-Walter De Neve, Sebastian Fehrler, Robert 847

Huber, Michael Grimm, Lennart Kaplan, Stefan Klonner, Olexiy Kyrychenko, Jorg 848

Peters, Massimo Phillipini, Atonu Rabbani, Alfonso Sanchez, Kirk R. Smith and 849

Johannes Urpelainen for their insightful comments that have helped us to improve the 850

draft of this paper.].

i) We note that you have provided funding information that is not currently declared in your Funding Statement. However, funding information should not appear in the Acknowledgments section or other areas of your manuscript. We will only publish funding information present in the Funding Statement section of the online submission form.

ii) Please remove any funding-related text from the manuscript and let us know how you would like to update your Funding Statement. Currently, your Funding Statement reads as follows:

 [The project benefitted from a scholar exchange grant provided to K.M. and P.D. by the Indo-Swiss Joint Research Programme in the Social Sciences (Call 2015) jointly funded by the Indian Council for Social Science Research and the Swiss State Secretariat for Education, Research and Innovation (https://www.unil.ch/international/home/menuguid/evenements/archives/indo-swiss-programme.html).

The funders had no role in study design, data collection and analysis, decision to publish, or preparation of the manuscript.].

iii) Additionally, because some of your funding information pertains to commercial funding, we ask you to provide an updated Competing Interests statement, declaring all sources of commercial funding.

iv) In your Competing Interests statement, please confirm that your commercial funding does not alter your adherence to PLOS ONE Editorial policies and criteria by including the following statement: "This does not alter our adherence to PLOS ONE policies on sharing data and materials.” as detailed online in our guide for authors  http://journals.plos.org/plosone/s/competing-interests.  If this statement is not true and your adherence to PLOS policies on sharing data and materials is altered, please explain how.

v) Please include the updated Competing Interests Statement and Funding Statement in your cover letter. We will change the online submission form on your behalf.

Additional Editor Comments (if provided):

Two expert reviewers have identified various shortcomings in the present version of the manuscript. These should be addressed fully in a revised version for potential publication.

Reviewers' comments:

Reviewer's Responses to Questions

**Comments to the Author**

1. Is the manuscript technically sound, and do the data support the conclusions?

Reviewer #1: Partly

Reviewer #2: Partly

2. Has the statistical analysis been performed appropriately and rigorously? 

Reviewer #1: Yes

Reviewer #2: Yes

3. Have the authors made all data underlying the findings in their manuscript fully available?

Reviewer #1: Yes

Reviewer #2: Yes

4. Is the manuscript presented in an intelligible fashion and written in standard English?

Reviewer #1: Yes

Reviewer #2: Yes

5. Review Comments to the Author

Reviewer #1: This is an interesting paper that will add new evidence to the domain literature. However, I would like to see some major and minor revisions to recommend it for publication. The comments are available in the attached report file.

Reviewer #2: This paper explores the effect of providing health-related information on the choice of energy source for cooking. It uses an experimental design and compares WTP and actual LPG use as outcomes of interest that may (and did) vary between treatment and control groups. This is highly relevant both in terms of making the energy sector more efficient and mitigating avoidable premature deaths. To the extent that it touches big themes (energy and health), I would like to see the paper motivated in a stronger way by linking these themes to the sustainable development goals.

Regarding the information in the first paragraph of the introduction, if global premature death associated with health problems of using biomass cooking is 4 million, that of India’s being 1 million sounds too much. I didn’t do fact check but it doesn’t sound right: Either the former has to be bigger of the latter smaller, or the figures were observed at different times.

The information in paragraph 4 (of the introduction) comes from the survey, but it is being used to justify why information intervention may induce change in behavior. I think the introduction can be structured in a better way. At this point, readers would expect to see the gap, research questions, hypotheses, contribution of the study etc.

Regarding the treatment, I don’t know how a poor person who can hardly afford it will change behavior and buy more only because they have enough information. Where will they get the money from? In the case of loan (line 130, page 4), they may also be evading repayment by not refilling. I think the central issue is not information but economics (which the authors highlight in different parts of the paper, e.g., lines 113-15).

I don’t see the relevance of the information provided in the paragraph that starts on line 89. The next paragraph (lie 94) reads: “Willingness to pay for improved…”. The question that comes to mind is where?

The results show that health awareness increased from 13% to 48%. I think 48% is too low and it has implications on the reliability of the design. As it is an experiment, the participants should understand the protocol properly. As much as the authors see the 48% as evidence for improved health awareness, a case can be made about the 52% being evidence for low internal validity of the experiment. If every participant understood the message, it would be reflected in a sweeping percentage being health aware.

When it comes to behavior elicitation, knowledge is important but experience may be vital. The authors could have asked if respondents had any serious health problems recently. It is possible that availability bias may cause differences in WTP.

Justification for the use of linear probability model (not probit/logit) is not strong enough. Yes, it is easier for interpretation but it would be better to explicitly state the gains and loses of making this choice.

On writing: instead of saying “our study doesn’t lend itself…” (line 37, page 2), it is clearer to be direct and say “our study does not measure…). This is just an example, and the paper can be written more succinctly and precisely.

Repetitions: Lines 64 and 67; paragraph on line 741.

Why are the number of households 550 but the number of individuals in the study 554?

6. PLOS authors have the option to publish the peer review history of their article (what does this mean?). If published, this will include your full peer review and any attached files.

Reviewer #1: No

Reviewer #2: No

---

## [Author Response · Author response to Decision Letter 0]

12 Feb 2020

See response letter to the reviewers attached.

---

## [Editor Report · Decision Letter 1]

18 Feb 2020

PONE-D-19-28805R1

Health awareness and the transition towards clean cooking fuels: Evidence from Rajasthan

PLOS ONE

Dear Prof. Michaelowa,

Thank you for submitting your manuscript to PLOS ONE. After careful consideration, we feel that it has merit but does not fully meet PLOS ONE’s publication criteria as it currently stands. Therefore, we invite you to submit a revised version of the manuscript that addresses the points raised during the review process.

The authors must address the outstanding issue regarding modeling and corresponding results for their experiment.

We would appreciate receiving your revised manuscript by Apr 03 2020 11:59PM. To enhance the reproducibility of your results, we recommend that if applicable you deposit your laboratory protocols in protocols.io, where a protocol can be assigned its own identifier (DOI) such that it can be cited independently in the future. For instructions see: http://journals.plos.org/plosone/s/submission-guidelines#loc-laboratory-protocols

We look forward to receiving your revised manuscript.

Kind regards,

Francisco X Aguilar

Academic Editor

PLOS ONE

Additional Editor Comments (if provided):

The authors have fully addressed all comments raised by reviewers with the exemption of one that merits correction.

First, theoretically the use of a probit/logit model is the correct specification for a probability study. In fact, the use of a logit model makes interpretation much easier by simple exponentiation of beta coefficients in the logistic specification. Empirically, the authors themselves report that running a probit model (it will be the same result for logit), they obtain similar results that "are even more precise". Why are the authors choosing to present the less precise (poorer-goodness-of-fit) results in the main body of the paper and have the more precise ones buried in a Supplementary file?

Beyond correctness in modeling and accuracy in empirical estimation, running a logistic regression is simple to do in any standard statistical software package. The interpretation of exponentiated beta logit coefficients is very straightforward.

The authors should remove the current results from the' linear probability' model and replace them accordingly with those from a logistic regression.

---

## [Author Response · Author response to Decision Letter 1]

3 Apr 2020

PONE-D-19-28805R1

Health awareness and the transition towards clean cooking fuels: Evidence from Rajasthan

PLOS ONE

Response to the reviewer

We are happy to hear that our revisions have fully addressed all comments, except some further changes you would like regarding the statistical modelling and the corresponding presentation of our results.

The issues raised were the following: 

First, theoretically the use of a probit/logit model is the correct specification for a probability study. In fact, the use of a logit model makes interpretation much easier by simple exponentiation of beta coefficients in the logistic specification. Empirically, the authors themselves report that running a probit model (it will be the same result for logit), they obtain similar results that "are even more precise". Why are the authors choosing to present the less precise (poorer-goodness-of-fit) results in the main body of the paper and have the more precise ones buried in a Supplementary file? Beyond correctness in modeling and accuracy in empirical estimation, running a logistic regression is simple to do in any standard statistical software package. The interpretation of exponentiated beta logit coefficients is very straightforward. The authors should remove the current results from the' linear probability' model and replace them accordingly with those from a logistic regression.

We have now revised the econometric modelling and the presentation in the text accordingly. 

Throughout the paper, we now use logit models to present the results for estimations with binary outcome variables. We keep the code for linear probability models for comparison in Appendix S6, in case the readers are interested in checking the robustness of our results to the modelling approach. 

The specific adjustments are as follows: 

• Section “Effect of health messaging on voucher use among voucher owners”:

We present the results in terms of odds ratios in Table 3 and in terms of predicted probabilities in Table 4. The corresponding discussion of results (lines 623-698) is adjusted accordingly.

• Section “Joint effect of health messaging on voucher use”:

We present the results in terms of odds ratios in Table 5 and in terms of predicted probabilities in Table 6. The corresponding discussion of results (lines 700-739) is adjusted accordingly.

• Section “Testing the information channel”:

In Table 7. “Treatment effect on health-awareness”, Col. 1 we now show average marginal effects based on a logit model. We do not show odds ratios here because the two other models in this table also show marginal effects as they relate to linear models for non-binary dependent variables. This facilitates the comparison across columns. 

The discussion remains essentially unchanged because the average marginal effects from the logit model are almost completely identical with the coefficient estimates of the linear probability model previously shown in Col. 1 of this table.

• Appendix S5, Section “Offer Prices and voucher use probability:” For the regression including an explicit modelling of the role of offer prices explicitly, we now use a logistic model, too. Thus, we present our results in Table 1 in terms of odds ratios, and the graphical illustration of changes in the predicted probabilities in Figure 2 is based on the logit model, too. The corresponding discussion of results has been adjusted and expanded to explain the rather tricky interpretation of the interaction between the discount and the health messaging. As we explain there, the graphical presentation as well as the presentation in terms of changes in predicted probabilities appear most instructive in this context. This discussion appears important to us given its relevance for policy making. (Only the results of this discussion are mentioned in the main document, see lines 737-740). 

• Abstract and conclusion: Due to marginal changes in our results small text adjustments were required regarding the comparison of a price reducing policy (discount) and health messaging (see 2nd last sentence in the Abstract, and Conclusion, lines 837-838).

Please note that the changes mentioned above (i.e., the changes in response to the review decision of 18 February 2020) are highlighted in yellow color in the Word document in order to distinguish them from prior changes in response to the first round of reviewer comments. All changes to the originally submitted version (from December 2019) are still visible in the track-changes mode.

---

## [Editor Report · Decision Letter 2]

6 Apr 2020

Health awareness and the transition towards clean cooking fuels: Evidence from Rajasthan

PONE-D-19-28805R2

Dear Dr. Michaelowa,

We are pleased to inform you that your manuscript has been judged scientifically suitable for publication and will be formally accepted for publication once it complies with all outstanding technical requirements.

With kind regards,

Francisco X Aguilar

Academic Editor

PLOS ONE

---

## [Editor Report · Acceptance letter]

8 Apr 2020

PONE-D-19-28805R2 

Health awareness and the transition towards clean cooking fuels: Evidence from Rajasthan 

Dear Dr. Michaelowa:

I am pleased to inform you that your manuscript has been deemed suitable for publication in PLOS ONE. Congratulations! Your manuscript is now with our production department. 

With kind regards,

on behalf of

Dr. Francisco X Aguilar 

Academic Editor

PLOS ONE